# Building inclusive rehabilitation education: A scoping review protocol on EDI approaches and impact

Kristina M. Kokorelias[1,2,3,4]*, Vicky Chau[3,5], Helana Marie Boutros[6,7], Sachindri Wijekoon[8], Hardeep Singh[1,2], Michael Kalu[9], Maurita T. Harris[7]

1 Department of Occupational Science & Occupational Therapy, Temerty Faculty of Medicine, University of Toronto, Toronto, Canada, 2 Toronto Rehabilitation Institute, University Health Network, Toronto, Ontario, Canada, 3 Section of Geriatrics, Department of Medicine, Sinai Health System and University Health Network, Toronto, Canada, 4 Temerty Faculty of Medicine, Rehabilitation Sciences Institute , University of Toronto, Toronto, Canada, 5 Division of Geriatric Medicine, Temerty Faculty of Medicine, University of Toronto, Toronto, Canada, 6 Department of Health, Faculty of Social Sciences, Society and Aging, McMaster University, Hamilton, Canada, 7 Faculty of Liberal Arts, Wilfred Laurier University, Brantford, Canada, 8 School of Occupational Therapy, Western University, London, Canada, 9 School of Kinesiology and Health Science, York University, Toronto, Canada,

* k.kokorelias@utoronto.ca

## Abstract

### Introduction

Equity, diversity, and inclusion (EDI) interventions are critical in fostering accessible and supportive educational environments in rehabilitation professions such as physiotherapy, occupational therapy, and speech-language pathology. These interventions promote cultural competence and reduce systemic inequities, enabling rehabilitation providers to address healthcare disparities and meet the diverse needs of patient populations. Despite increasing awareness, the scope, implementation, and impact of EDI interventions in rehabilitation education remain underexplored.

### Aim

This scoping review aims to map the existing literature on EDI interventions in rehabilitation education, focusing on methods, strategies, and their impact on learners' outcomes, including cultural competence, professional practice, and attitudes toward diversity.

### Methods

The review follows the Joanna Briggs Institute methodology and PRISMA-Scr guidelines. Studies involving healthcare students and educators in rehabilitation disciplines are eligible. The concept includes EDI-focused interventions, such as curricular reforms, diversity training, and inclusive teaching practices. Eligible contexts include academic and healthcare settings. A comprehensive search strategy will be applied

**Data availability statement:** No datasets were generated or analysed during the current study. All relevant data from this study will be made available upon study completion.

**Funding:** The author(s) received no specific funding for this work.

**Competing interests:** The authors have declared that no competing interests exist.

across databases including MEDLINE, EMBASE, and CINAHL, supplemented by grey literature searches. Inclusion criteria follow the Population, Concept, and Context (PCC) framework.

## Results

This review will chart interventions, their educational context, and their outcomes. Key data to be extracted include study design, intervention details, and learner outcomes, emphasizing cultural competence and attitudes.

## Conclusion

Findings will provide a consolidated framework for developing evidence-based EDI interventions in rehabilitation education, addressing gaps in preparing a workforce capable of delivering equitable care in diverse contexts. Recommendations will guide future research and practice, advancing inclusivity and equity in rehabilitation education.

## Introduction

Equity, diversity, and inclusion (EDI) interventions are critical in fostering an educational environment that is accessible, equitable, and supportive of diverse individuals in the rehabilitation professions [1]. The rehabilitation professions, including physiotherapy, occupational therapy, and speech-language pathology, are increasingly required to adopt inclusive practices that reflect the diversity of the populations they serve [2,3]. Given the diverse cultural, socio-economic, and personal backgrounds of providers, students and professionals in these fields, it is critical to develop and implement strategies that address inequities and enhance the learning experience for all individuals. As such, EDI training interventions help future rehabilitation providers not only to recognize and address their own biases, but also to appreciate the complexities of providing equitable care to individuals from underrepresented or marginalized groups [4,5]. We use the term EDI interventions to refer to deliberate strategies and actions designed to promote fairness, representation, and an inclusive environment across different social, cultural, and demographic groups [6]. These interventions aim to address and mitigate systemic inequalities by ensuring that all individuals have equal access to opportunities, resources, and support regardless of their background or identity. In the context of rehabilitation education, EDI strategies and practices may involve incorporating inclusive teaching practices, promoting diverse perspectives, ensuring representation in curricula and pedagogy and implementing policies that reduce barriers to access appropriate rehabilitation services [5].

Integrating EDI practices into rehabilitation education to equip students and rehabilitation professionals is particularly important, given the growing recognition that healthcare disparities exist across various patient populations [7]. These disparities are often driven by factors such as race, ethnicity, gender, age, disability,

and socioeconomic status, all of which can affect health outcomes and access to care [5]. Unlike medicine, which often focuses on diagnosing and treating diseases or conditions, rehabilitation is centered around improving the quality of life and enhancing the functional independence of patients with diverse abilities and backgrounds. [8]. This requires rehabilitation professionals to engage with patients on a clinical and personal level [9], recognizing the social, cultural, and emotional factors that affect recovery and well-being [10,11]. Therefore, EDI strategies in rehabilitation education must emphasize cultural competence, sensitivity to diverse lived experiences, and the ability to adapt interventions to various social, physical, and psychological contexts. [12] These strategies go beyond addressing disparities in healthcare access, as seen in medical education, by focusing on fostering relationships that respect patient autonomy and individuality within the context of their specific needs. As a result, EDI strategies in rehabilitation education must be tailored to develop clinical skills and interpersonal and communication skills that are critical for working effectively with diverse populations [13].

Educational strategies aimed at promoting EDI in these fields have gained significant attention, particularly in response to the global call for reform in higher education and healthcare [6]. Despite growing awareness, there remains limited systematic evidence on the scope, implementation, and impact of such interventions within the rehabilitation disciplines. To date, various approaches have been trialed, ranging from curricular changes [14,15] to institutional reforms and training programs to enhance cultural competency and reduce bias [16,17]. However, the effectiveness of these interventions and the specific educational strategies employed have not been thoroughly examined in a consolidated framework.

This scoping review seeks to map and analyze the existing literature on EDI training interventions and the impact of those interventions within rehabilitation education. By identifying the range of strategies employed and evaluating their impact on learners' outcomes, this review aims to inform future developments in EDI training programs to equip students to treat diverse patient populations, ensuring they are evidence-based and tailored to meet the needs of diverse student populations. Ultimately, the findings will contribute to the ongoing efforts to create a more inclusive and equitable rehabilitation workforce equipped to address the healthcare challenges of an increasingly diverse society.

## Aim

The objective of this paper is to outline a protocol for a scoping review of the existing literature on effective strategies for engaging patient partners in the research process. This investigation will explore a) the methods, interventions and approaches for incorporating EDI practices into rehabilitation healthcare education; and b) the effects of such programs on learner outcomes.

## Review objectives

The objectives of this scoping review are: (1) To identify and describe the methods, interventions, and approaches used to incorporate EDI principles into rehabilitation healthcare education; (2) To evaluate the impact of these EDI programs on learners' outcomes, including cultural competence, student attitudes, and professional practice.

Our definition of EDI encompasses strategies aimed at reducing disparities, fostering inclusive learning environments, and addressing systemic barriers in healthcare education. Additionally, "rehabilitation education" refers broadly to training and curricula designed for healthcare professionals in fields such as physiotherapy, occupational therapy, and speStakeholder Involvementech-language pathology, among others.

## Methods

This scoping review will be conducted in adherence to the Joanna Briggs Institute (JBI) methodology for scoping reviews [18] and the PRISMA-P guidelines for protocol development[19], following a comprehensive and systematic approach to mapping the existing literature on EDI) practices in rehabilitation healthcare education. The goal is to identify the breadth and range of methods, interventions, and approaches used to incorporate EDI principles into rehabilitation education and to assess their effects on learners' outcomes.

The idea for this scoping review emerged in response to requests from in-patient rehabilitation providers and educators who are closely connected to our research team. These providers expressed concerns about the persistent gaps in EDI) within rehabilitation healthcare education. They specifically highlighted the need for evidence-informed strategies to integrate EDI principles into their training programs better. Given the growing recognition of the importance of cultural competence and inclusive professional practice in healthcare, these stakeholders emphasized the urgency of identifying effective methods and interventions to improve student preparedness in diverse clinical settings. Their feedback directly informed our decision to conduct a comprehensive review to understand current approaches and evaluate their impact on learner outcomes. These individuals will actively contribute to our team throughout the scoping review process, ensuring the co-creation of knowledge aligned with guidance from the JBI Scoping Review Methodology Group [20]. By engaging these rehabilitation providers as knowledge users, we aim to align our findings with real-world educational needs and enhance the applicability of our recommendations.

**Eligibility criteria**

The eligibility criteria for inclusion in this review will follow the Population, Concept, and Context (PCC) framework outlined by JBI [21]. Studies will be included based on the following criteria:

**Population.**

- Healthcare education students, including those in rehabilitation professions such as physiotherapy, occupational therapy, speech-language pathology, and rehabilitation counseling (including team-based education whereby rehabilitation professions are considered)

- Educators involved in the delivery of EDI interventions in rehabilitation education.

**Concept.**

- Interventions, strategies, or approaches focused on integrating EDI principles into rehabilitation education. This includes but is not limited to curriculum reforms, workshops, diversity training, simulation-based learning, and community engagement programs.

- Studies assess students' cultural competence, sensitivity to diversity, changes in attitudes toward diverse populations, and professional practice behaviour.

**Context.**

- Studies conducted in or with academic institutions offering rehabilitation healthcare programs, healthcare organizations, and community settings.

- Articles published in peer-reviewed journals, grey literature, and government or institutional reports related to rehabilitation education. There are no restrictions on geographical location or publication date.

The scope will cover comparative studies (e.g., randomized, controlled, cohort, quasi-experimental) and non-comparative approaches (e.g., surveys, narratives, case studies), as well as reports, policy documents, and educational materials related to EDI practices in academic and healthcare settings. We will exclude studies not focusing on EDI in rehabilitation healthcare education and research on other healthcare professions (e.g., medicine, nursing) that do not specifically address rehabilitation fields.

**Information sources**

The comprehensive search strategy will be developed by a health sciences librarian (CDC) and the research team, with input from rehabilitation experts. The search strategy will be first developed in Medline and validated by retrieving a key set of relevant studies and peer-reviewed using the PRESS strategy [22]. Searches will not be limited by a time

period. Keywords will include *Equity, Diversity, Inclusion, Cultural Competence, Rehabilitation Education, Healthcare Education, Curriculum Development, Cultural Sensitivity, Bias in Education, Inclusive Teaching,* and specific rehabilitation professions such as *Physiotherapy Education, Occupational Therapy Education, Speech-Language Pathology Education.* Boolean operators (AND, OR) will combine these terms. The search will be limited to studies published in English, but no date restrictions will be applied. The MEDLINE search strategy will be translated into EMBASE, PubMed, CINAHL, ERIC, Scopus, and PsycINFO. Reference lists of included studies will be searched for citations. We will also conduct a hand search of key journals and publications focusing on rehabilitation education and EDI, such as *Disability and Rehabilitation, Journal of Rehabilitation Research and Development, Journal of Allied Health, and Medical Education.*

We will conduct a comprehensive search of grey literature to uncover non-indexed sources, such as dissertations, government reports, conference materials, practice guidelines, and educational resources. This search will utilize databases like Open Grey, Conference Proceedings Index, ProQuest Dissertations and Theses, and Google. Given resource limitations, the grey literature search will be restricted to Canada and the United States to maintain feasibility. This geographic focus will allow for a more efficient allocation of resources, ensuring a thorough review within these regions.

We will collect and store the bibliographic details of all included sources using EndNote [23]. After removing duplicates, the references will be imported into Covidence for further processing without any limitations on language, date, or study type [24].

### Study selection

The study selection process will follow the PRISMA-Scoping Review Extension guidelines (PRISMA-SCr) [25]. A two-step screening process will be employed. Two reviewers will independently assess the titles and abstracts according to the selection criteria, categorizing them as 'include,' 'exclude,' or 'uncertain.' Any disagreements will be addressed through discussion between the reviewers, with a third reviewer consulted if consensus cannot be reached. For studies marked as 'included' or 'maybe,' full-text articles will be retrieved and reviewed independently in duplicate to confirm eligibility. The same two independent reviewers will review full-text articles of potentially eligible studies. Studies that meet the inclusion criteria will be retained for data extraction. If there is disagreement at this stage, the third reviewer (MTH) will adjudicate. Throughout all phases inter-rater reliability will be measured using Cohen's kappa [26].

As with the academic literature, two independent reviewers will assess grey literature sources, and inter-rater reliability will be measured using Cohen's kappa [26].

### Data extraction

Two reviewers will independently review and chart all included studies using a pilot-tested data abstraction form. Charting will involve organizing, categorizing, and interpreting the data based on key themes and issues. A pilot test will be conducted on approximately four articles to ensure consistency and refine the data abstraction process. Any necessary adjustments will be made and shared with the team before extracting data from the remaining articles.

The following data will be extracted when available (subject to change depending on the included studies): study citation, publication type (published or unpublished), study design (e.g., quantitative, qualitative, mixed methods), study context (e.g., location, healthcare setting), intervention details (e.g., goal-setting process, EDI dimensions addressed, outcomes), and the educational context (e.g., undergraduate, graduate, clinical training). Discrepancies in data extraction will be resolved through discussion, and in case of missing data or uncertainties, the primary authors will be contacted for clarification. However, in line with the scoping review methodology, we expect the data charting process to be iterative, evolving based on the literature identified [27]. A study quality assessment or meta-analysis of quantitative results will not be performed, as these are not within the scope or objectives of this review. Instead, all included studies will be synthesized.

To enhance the rigor and generalizability of our findings, we will apply the Mixed Methods Appraisal Tool (MMAT) to assess the quality of included studies, particularly those employing mixed-methods approaches [28]. This tool will allow for a systematic evaluation of methodological quality across qualitative, quantitative, and mixed-methods studies, strengthening the reliability of our synthesis [28].

## Data analysis and synthesis

A narrative synthesis approach will be employed to analyze the extracted data, providing a detailed overview of the methods, strategies, and interventions identified across the studies. This synthesis will primarily summarize the key findings related to integrating EDI practices into rehabilitation education. It will focus on how these practices are applied within educational settings and their impact on learner outcomes. Given the nature of scoping reviews, a formal quality assessment of the included studies will not be conducted [27]. However, a brief commentary on the methodological rigor of the included studies will be provided, noting any potential limitations or biases that may have affected the interpretation of findings. Through a systematic review of the literature, we will identify recurring themes and patterns in implementing EDI strategies, noting the various ways these interventions are designed and executed.

The narrative synthesis will also highlight variations in approaches, evaluating how different methods contribute to the diverse outcomes observed across studies. This process will allow for an exploration of the contextual factors—such as the setting, participant demographics, and cultural contexts—that may influence the effectiveness of EDI interventions. A thematic analysis will be conducted to categorize and critically analyze the interventions, identifying the key components of each approach and assessing their reported effectiveness.

Where feasible, the results will be grouped and analyzed according to specific types of interventions, such as curriculum-based reforms, experiential learning opportunities, and educator training programs. This categorization will enable us to better understand how different EDI strategies contribute to various educational outcomes, including cultural competence improvements, student attitude shifts, and professional behavior changes. In addition to qualitative synthesis, we will provide frequency counts for each type of intervention and outcome, offering a clearer view of the prevalence and commonality of different EDI practices across the studies. By drawing connections between intervention types and their associated impacts and documenting the frequency with which each approach is applied, this analysis will provide a clearer understanding of which approaches are most effective in integrating EDI principles into rehabilitation education, and offer guidance for future educational practices in this field.

## Stakeholder involvement

In our study, we are adopting an integrated knowledge translation approach by actively engaging knowledge users throughout all stages of the research process [29]. Our team includes two key knowledge users who assisted with the research objectives and study methodology. However, as the review continues, their involvement will ensure that our findings are aligned with the needs of rehabilitation professionals and policymakers, and they will offer valuable input at various stages, from suggesting additional information sources to leading knowledge translation activities at the end of the study. In addition to involving our core knowledge users, we will extend our consultation to other rehabilitation educators and practitioners to ensure broad perspectives are considered. We will also recruit additional shareholders, including rehabilitation educators, students, and patient partners, to be consulted to ensure the relevance and applicability of the findings. Input from stakeholders will be sought during the interpretation phase to inform the implications of the review's findings for practice, policy, and future research directions. This collaborative effort will help us refine our results and enhance the relevance of the research.

Initially, we will reach out to rehabilitation educators, students, and patient partners through targeted recruitment strategies, such as leveraging professional networks, academic institutions, and patient advocacy groups. To engage potential

participants, we will use email invitations, social media platforms, and direct outreach at relevant conferences or events. Our recruitment materials will clearly outline the purpose of the study, the expected time commitment, and the benefits of involvement.

Once recruited, stakeholders will be engaged through a series of virtual and in-person meetings, scheduled throughout the study. These meetings will provide an opportunity for stakeholders to review key findings, ask questions, and provide feedback. We will ensure that the meetings are structured to facilitate meaningful dialogue, with clear agendas, and will offer flexible scheduling to accommodate diverse availability. Stakeholders will also have the option to participate in smaller focus groups or one-on-one consultations, depending on the review stage. Regular check-ins will be scheduled at key milestones of the study, particularly during the data interpretation phase, to ensure ongoing engagement and to gather their insights on the implications of the findings for practice, policy, and future research directions. While the community of practice will provide input on knowledge dissemination strategies, we acknowledge the importance of ensuring methodological replicability. Therefore, stakeholder engagement will not be used to introduce new data sources beyond those identified through our systematic search strategy. Instead, their role will focus on refining interpretations and ensuring that findings are presented in ways that maximize impact and applicability for rehabilitation education stakeholders.

In addition to meetings, we will use online collaboration tools (e.g., shared documents and surveys) to gather input asynchronously, providing stakeholders with opportunities to contribute in a manner that best suits their schedules. This multi-method approach will ensure continuous engagement and a thorough understanding of the findings from multiple perspectives.

### Ethics and dissemination

Ethical approval is not required since this scoping review involves the analysis of existing literature and does not involve direct interaction with human participants. Stakeholder involvement in this study is limited to consultation on knowledge dissemination strategies and interpretation of findings, rather than participation in data collection or research activities. As such, this engagement does not meet the threshold for requiring ethics approval. The scoping review results will be disseminated through academic journals and presentations at conferences (e.g., The International Conference on Aging, Innovation & Rehabilitation). A summary of the findings will also be shared with stakeholders, including educators and rehabilitation professionals, to inform future training practices and improve EDI integration in rehabilitation education.

The scoping review is expected to take approximately six-twelve months from the initial search to final report submission.

## Results

We anticipate identifying a variety of interventions, such as curriculum reforms, workshops, diversity training, simulation-based learning, and community engagement programs, that have been used to address systemic inequities within these educational contexts. Additionally, we expect to find a range of outcomes related to learner development, including enhanced cultural competence, changes in attitudes toward diverse populations, and improved professional practice behaviors. The review will also report on gaps in the current literature, particularly regarding the specific effects of these interventions on long-term career outcomes and how these interventions may influence the quality of care provided to diverse patient populations.

## Discussion

This article aimed to develop and evaluate a protocol aimed at understanding the feasibility and effectiveness of a new intervention targeting healthcare professionals and primary health care systems. Given the pressing need for evidence-based strategies to improve care delivery, particularly for older adults with complex health conditions, this study

is designed to fill existing gaps in the literature regarding the implementation of such interventions. The forthcoming review is intended to guide future research in evaluating intervention outcomes, inform the development of best practices, and provide evidence on how these interventions can be scaled to address systemic challenges within health systems, such as inefficiency and fragmented care.

From an educational perspective, this review will have significant implications for curriculum design and the professional development of healthcare providers in rehabilitation fields. It will provide insights into the content, delivery, and effectiveness of EDI-focused training programs, helping to shape educational strategies that promote inclusivity and cultural competence. Additionally, the review may suggest ways better to integrate EDI principles into ongoing training and certification programs, ensuring rehabilitation professionals have the skills and knowledge necessary to deliver equitable care. By embedding EDI-focused education into rehabilitation training, the findings of this review will contribute to fostering a more inclusive and effective healthcare workforce.

The research implications of this review are considerable. By synthesizing evidence on EDI education interventions, this study will identify effective strategies and areas where further research is needed. The findings will provide a clearer understanding of how EDI training in rehabilitation settings influences not only clinicians' attitudes and competencies but also the quality of care delivered to underrepresented or marginalized patient groups. It will also help identify best practices in designing and implementing EDI-focused educational programs and evaluate how they can be scaled or adapted across different healthcare environments and cultural contexts.

The clinical practice implications of this review are profound. By determining the most effective EDI education interventions, this study will offer evidence-based recommendations for integrating EDI principles into clinical practice in rehabilitation settings. Healthcare professionals will be better equipped to recognize and address the unique needs of diverse populations, reduce biases in treatment, and foster a more inclusive care environment. The findings may also inform the development of clinical guidelines and training modules to promote equitable care practices, ultimately improving health outcomes for patients from marginalized or underrepresented groups.

## Limitations

Several limitations must be considered in this scoping review protocol. First, the focus on gray literature conducted in Canada and the United States may limit the generalizability of the findings to other regions where rehabilitation education and healthcare practices may differ. Additionally, relying on English-language publications and excluding non-peer-reviewed sources outside these countries could introduce a bias, particularly in areas with differing educational and healthcare systems. Another limitation is the heterogeneity of the studies included in this review. The inclusion of a wide variety of study designs, such as qualitative, quantitative, and mixed-methods approaches, may make it challenging to compare outcomes across interventions. Furthermore, the broad range of EDI interventions and measured learner outcomes may complicate the findings' synthesis. We acknowledge the limitations of using policy documents and reports that may not explicitly assess the impacts of DEI interventions within research environments. However, these sources provide valuable contextual information on broader structural and policy-level influences that shape inclusive rehabilitation education. To mitigate potential biases and confounding variables, we will critically assess these documents using established frameworks for evaluating policy impact and evidence quality.

## Conclusion

This scoping review will provide valuable insights into the landscape of EDI interventions in rehabilitation healthcare education. Mapping the methods and outcomes of existing programs will contribute to developing evidence-based practices for integrating EDI principles into rehabilitation curricula. The review will emphasize the need for continued innovation in educational strategies to address systemic inequities and promote culturally competent, inclusive care. Ultimately, the findings of this review will guide the design of future EDI interventions, inform policy development in rehabilitation

education, and help ensure that the next generation of rehabilitation professionals is equipped to meet the diverse needs of patients in an increasingly multicultural society.

## Supporting information

**S1= PRISMA P Checklist.**
(PDF)

**S2 Appendix.**
(DOCX)

## Author contributions

**Conceptualization:** Kristina M Kokorelias, Helana Marie Boutros, Hardeep Singh, Michael Kalu, Maurita T Harris.

**Data curation:** Kristina M Kokorelias, Maurita T Harris.

**Formal analysis:** Kristina M Kokorelias.

**Funding acquisition:** Kristina M Kokorelias.

**Investigation:** Kristina M Kokorelias, Maurita T Harris.

**Methodology:** Kristina M Kokorelias, Vicky Chau, Maurita T Harris.

**Project administration:** Kristina M Kokorelias, Maurita T Harris.

**Resources:** Kristina M Kokorelias, Maurita T Harris.

**Software:** Kristina M Kokorelias.

**Writing – original draft:** Kristina M Kokorelias.

**Writing – review & editing:** Vicky Chau, Helana Marie Boutros, Sachindri Wijekoon, Maurita T Harris.

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
