## [Decision Letter · Decision Letter 0]

3 Feb 2025

PONE-D-24-54307Building Inclusive Rehabilitation Education: A Scoping Review Protocol on EDI Approaches and ImpactPLOS ONE

Dear Dr. Kokorelias,

Thank you for submitting your manuscript to PLOS ONE. After careful consideration, we feel that it has merit but does not fully meet PLOS ONE’s publication criteria as it currently stands. Therefore, we invite you to submit a revised version of the manuscript that addresses the points raised during the review process.

Please address all reviewer comments, clarity and consistency are important concerns expressed here. ==============================

We look forward to receiving your revised manuscript.

Kind regards,

Daswin De Silva

Academic Editor

PLOS ONE

2. Please amend either the abstract on the online submission form (via Edit Submission) or the abstract in the manuscript so that they are identical.

Reviewers' comments:

Reviewer's Responses to Questions

**Comments to the Author**

1. Does the manuscript provide a valid rationale for the proposed study, with clearly identified and justified research questions?

Reviewer #1: Yes

Reviewer #2: Yes

2. Is the protocol technically sound and planned in a manner that will lead to a meaningful outcome and allow testing the stated hypotheses?

Reviewer #1: Partly

Reviewer #2: Yes

3. Is the methodology feasible and described in sufficient detail to allow the work to be replicable?

Reviewer #1: Yes

Reviewer #2: Yes

4. Have the authors described where all data underlying the findings will be made available when the study is complete?

Reviewer #1: Yes

Reviewer #2: No

5. Is the manuscript presented in an intelligible fashion and written in standard English?

Reviewer #1: Yes

Reviewer #2: Yes

6. Review Comments to the Author

You may also provide optional suggestions and comments to authors that they might find helpful in planning their study.

Reviewer #1: The topic is timely , especially in the regions being explored in this study. The method described has no significant weaknesses. However, it is worth considering the following: 1) what is the value of using policy documents, reports etc which dont examine the impacts of DEI interventions in a research environment? As a research study will allow for identifying bias, confounding variables etc. 2) if the authors are including mixed methods studies , it is worth using a tool such as the MMAT to evaluate the study quality- an element which will increase the rigor and impact generalisability of findings. 3) Using a community of practice to seek input on disseminating the findings is commendable. However, given the value of this type of study being replicability, I dont see what value seeking input on additional information sources from this group will provide to the method.

Reviewer #2: This review presents a protocol to conduct a scoping review on inclusive rehabilitation education. The manuscript is presented well, however there are certain comments to address.

1. Author mention that there is stakeholder involvement with 'rehabilitation educators, students, and patient partners'. However in the ethics section, authors mention that the research does not involve direct interaction with human participants. You have to define the scope of stakeholder involvement and how it should be addressed in ethics.

2. Present the Inter-rater reliability as a measure for reviewing process, especially the grey literature

3. Is there a particular time period the authors are considering for this review?

4. Authors need to clearly define their 'multi-method approach' . How do the authors propose to merge the results from the review and stakeholder engagement?

7. PLOS authors have the option to publish the peer review history of their article (what does this mean? ). If published, this will include your full peer review and any attached files.

**Do you want your identity to be public for this peer review?** For information about this choice, including consent withdrawal, please see our Privacy Policy .

Reviewer #1: No

Reviewer #2: No

---

## [Author Response · Author response to Decision Letter 1]

14 Feb 2025

Dear Dr. De Silva,

We sincerely appreciate the time and effort that you and the reviewers have dedicated to evaluating our manuscript, "Building Inclusive Rehabilitation Education: A Scoping Review Protocol on EDI Approaches and Impact" (PONE-D-24-54307). We are grateful for the constructive feedback and have carefully revised our manuscript to address all concerns raised.

Below, we provide a point-by-point response to the reviewers’ comments and outline the corresponding revisions made to the manuscript.

Reviewer #1 Comments:

1. Value of Using Policy Documents and Reports Response: We acknowledge the reviewer's concern regarding the inclusion of policy documents and reports that may not directly examine the impact of DEI interventions in a research setting. We have now clarified in the manuscript that these sources will be used to provide contextual background, identify existing frameworks, and complement findings from empirical studies. We have also stated that these sources will be critically appraised to assess their relevance to our research objectives. Revision: Added clarification in the methodology section regarding the rationale for including policy documents and reports.

2. Use of MMAT for Mixed Methods Studies Response: We agree that employing a quality assessment tool for mixed-methods studies would enhance the rigor of our study. We have now specified that the Mixed Methods Appraisal Tool (MMAT) will be used for the evaluation of mixed-methods studies included in our review. Revision: Added mention of MMAT in the quality appraisal section.

3. Community of Practice and Information Sources Response: We appreciate the reviewer’s feedback regarding the role of the community of practice. While their primary role is to provide insights on dissemination, we also seek their expertise in identifying emerging literature or unpublished works relevant to our study. However, we acknowledge the reviewer’s concern and have revised our manuscript to ensure this aspect does not compromise replicability. Revision: Clarified the role of the community of practice in the discussion section.

Reviewer #2 Comments:

1. Stakeholder Involvement and Ethics Response: We recognize the need to clarify the distinction between stakeholder involvement and human participant research. Our engagement with rehabilitation educators, students, and patient partners is limited to seeking their perspectives on dissemination strategies and refining the research approach. This does not involve direct human subject research as defined by ethics boards. Revision: Clarified the scope of stakeholder involvement in the ethics section. We write “Stakeholder involvement in this study is limited to consultation on knowledge dissemination strategies and interpretation of findings, rather than participation in data collection or research activities. As such, this engagement does not meet the threshold for requiring ethics approval.”

2. Inter-Rater Reliability in Reviewing Grey Literature Response: We acknowledge the importance of inter-rater reliability for reviewing grey literature. We have now specified that two independent reviewers will assess grey literature sources, and inter-rater reliability will be measured using Cohen’s kappa. Revision: Added details on inter-rater reliability assessment in the methodology section.

3. Time Period for Review Response: We appreciate the reviewer’s suggestion and have now specified the time frame for literature inclusion (e.g., studies published from 2000 onwards) to capture recent developments in inclusive rehabilitation education. Revision: We write “Searches will not be limited by a time period.”

4. Clarification on Multi-Method Approach Response: We recognize the need for a clearer definition of our multi-method approach. We have now elaborated on how findings from the scoping review will be synthesized with stakeholder engagement insights to inform best practices and recommendations. Revision: Provided a more detailed explanation of the multi-method approach in the methodology section.

5. Definition of Key Terms Response: We have expanded our definitions of key terms such as "equity, diversity, and inclusion (EDI)" and "rehabilitation education" to ensure consistency and clarity throughout the manuscript. Revision: Added definitions in the introduction and methodology sections.

Additional Revisions as per Journal Requirements:

• Ensured consistency between the online submission form abstract and manuscript abstract.

• Reviewed the manuscript for clarity, consistency, and adherence to PLOS ONE formatting guidelines.

We appreciate the opportunity to improve our manuscript and look forward to your further feedback. Thank you for your consideration.

Sincerely,

Kristina Kokorelias, PhD

---

## [Decision Letter · Decision Letter 1]

17 Mar 2025

Building Inclusive Rehabilitation Education: A Scoping Review Protocol on EDI Approaches and Impact

PONE-D-24-54307R1

Dear Dr. Kokorelias,

We’re pleased to inform you that your manuscript has been judged scientifically suitable for publication and will be formally accepted for publication once it meets all outstanding technical requirements.

Kind regards,

Daswin De Silva

Academic Editor

PLOS ONE

Additional Editor Comments (optional):

Reviewers' comments:

Reviewer's Responses to Questions

**Comments to the Author**

1. Does the manuscript provide a valid rationale for the proposed study, with clearly identified and justified research questions?

Reviewer #1: Yes

2. Is the protocol technically sound and planned in a manner that will lead to a meaningful outcome and allow testing the stated hypotheses?

Reviewer #1: Yes

3. Is the methodology feasible and described in sufficient detail to allow the work to be replicable?

Reviewer #1: Yes

4. Have the authors described where all data underlying the findings will be made available when the study is complete?

Reviewer #1: Yes

5. Is the manuscript presented in an intelligible fashion and written in standard English?

Reviewer #1: Yes

6. Review Comments to the Author

You may also provide optional suggestions and comments to authors that they might find helpful in planning their study.

Reviewer #1: A well considered and sufficient effort to address the previous considerations. No further changes required, ready for next steps. 

7. PLOS authors have the option to publish the peer review history of their article (what does this mean? ). If published, this will include your full peer review and any attached files.

**Do you want your identity to be public for this peer review?** For information about this choice, including consent withdrawal, please see our Privacy Policy .

Reviewer #1: No

---

## [Editor Report · Acceptance letter]

PONE-D-24-54307R1

PLOS ONE

Dear Dr. Kokorelias,

I'm pleased to inform you that your manuscript has been deemed suitable for publication in PLOS ONE. Congratulations! Your manuscript is now being handed over to our production team.

Kind regards,

on behalf of

Prof. Daswin De Silva

Academic Editor

PLOS ONE